# Modeling the Interactions Between Chemicals and Proteins to Predict the Health Consequences of Air Pollution

**DOI:** 10.3390/ijerph22030418

**Published:** 2025-03-13

**Authors:** Md. Ramjan Sheikh, Hasna Heena Mahmud, Md. Saikat Hossen, Disha Saha, Md. Ekhlas Uddin, Md. Fuad Hossain, Md. Kamruzzaman Munshi, Abu Ali Ibn Sina

**Affiliations:** 1Department of Biochemistry and Molecular Biology, Gono Bishwabidyalay, Dhaka 1344, Bangladesh; rsheikh1996@universepg.com (M.R.S.); ekhlas.pgr20191@juniv.edu (M.E.U.); head.bmb@gonouniversity.edu.bd (M.F.H.); 2Department of Disaster Science and Climate Resilience, University of Dhaka, Dhaka 1000, Bangladesh; hasnaheenamahmud@gmail.com; 3Department of Soil and Environmental Sciences, University of Barishal, Barishal 8254, Bangladesh; msaikat22.ses@bu.ac.bd; 4Department of Crop Physiology and Ecology, Hajee Mohammad Danesh Science and Technology University, Dinajpur 5200, Bangladesh; dishasaha335@gmail.com; 5Institute of Food and Radiation Biology, Bangladesh Atomic Energy Commission, Dhaka 1000, Bangladesh; kzaman_munshi@yahoo.com; 6Center for Personalized Nanomedicine, Australian Institute for Bioengineering & Nanotechnology (AIBN), The University of Queensland, Brisbane, QLD 4072, Australia

**Keywords:** PPI, PCI, functional analysis, air pollution

## Abstract

The impacts of air pollution on human health have become a major concern, especially with rising greenhouse gas emissions and urban development. This study investigates the molecular mechanisms using the STITCH 4.0 and STRING 9.0 databases to analyze the interaction networks (PCI and PPI) associated with two air pollutants: carbon monoxide and hydrogen sulfide. The functional and pathway analysis related to these pollutants were performed by OmicsBox v.3.0. Additionally, critical proteins and their essential pathways were also identified by the Cytoscape networking tool v.3.10.3. AutoDock vina was employed to hypothetically determine the direct interactions of CO and H_2_S with the proteins that were found by STITCH. This study revealed that CO and H_2_S interacted with the different biological processes related to human health, including erythropoiesis, oxidative stress, energy production, amino acids metabolism, and multiple signaling pathways associated with respiratory, cardiovascular, and neurological functions. Six essential proteins were identified based on their degree of centrality, namely, FECH, HMOX1, ALB, CTH, CBS, and CBSL, which regulate various Reactome and KEGG pathways. Molecular docking analysis revealed that CO exhibited a strong interaction with ADI1, demonstrating a binding affinity of −1.9 kcal/mL. Alternately, the binding energy associated with the H_2_S interaction was notably weak (below −0.9 kcal/mL). This present research highlights the necessity for ongoing investigation into the molecular effects of air pollution to guide public health policies and interventions.

## 1. Introduction

Air pollution is a pervasive environmental issue that affects millions of people worldwide, contributing to a range of health problems [1]. The increased amount of greenhouse emissions has caused a significant deterioration of air quality, which is recognized as a growing concern, specifically due to its far-reaching implications for human health [2]. Due to industrialization and rapid urbanization, populations have faced multiple health consequences, such as respiratory, cardiovascular, and neurological diseases related to the increased prevalence of atmospheric pollutants [3]. These pollutants include, but are not limited to, particulate matter (PM), nitrogen dioxide (NO_2_), sulfur dioxide (SO₂), carbon monoxide (CO), ozone (O₃), and volatile organic compounds (VOCs) [4]. However, extensive epidemiological research concerning these health consequences and their biomolecular basis has remained insufficiently understood [5]. Therefore, there is an urgent need for studies that examine the molecular and cellular mechanisms that lead to the negative effects of air pollution on health [6]. In this regard, one of the promising strategies for understanding these mechanisms is the chemical–protein networks, which act as an integrated model of protein–protein interactions (PPIs) and reveal how pollutants can change the biological processes at the molecular level [7].

Integrating chemical–protein and protein–protein interaction networks into air pollution-related studies will enable the harmonic exploration of complex biochemical paths disrupted by pollutants. Interactions of human chemicals and proteins are known as chemical–protein interaction networks [8]. These networks give information on how pollutants can attach to proteins, altering their behavior, status, and metabolism and initiating pathological conditions. Particulate matter and heavy metals can induce oxidative stress by interacting with proteins within the antioxidant response network in the body. Mapping these interactions can help researchers anticipate health risks that could arise from different pollutants and outline key proteins that could be used as the basis for new biomarkers to quickly diagnose diseases caused by pollution [7].

However, protein–protein interaction (PPI) networks elucidate the effects of these chemically modified proteins on larger-scale biological processes. Proteins interact with each other to create a PPI network, and any disturbance in the network leads to cascading effects and various systemic physiological imbalances [9]. The STITCH and STRING databases predict which proteins are most susceptible to modification by air pollutants and how these modifications contribute to disease progression, particularly in the hematology, respiratory, and cardiovascular systems. Identifying specific proteins and pathways affected by air pollutants is essential for mitigating or reversing their detrimental health impacts. Cytoscape is a tool that identifies critical proteins based on degree centrality, and OmicsBox is a bioinformatics tool that analyzes the functions and pathways of proteins [10]. Various airborne pollutants have been reported to interfere with proteins that regulate important cellular processes, including inflammation, immune response, and cellular metabolism [11]. These disturbances may result in chronic diseases, including asthma, chronic obstructive pulmonary disease (COPD), and cardiovascular disorders. Investigating PPI networks allows researchers to follow the chain reaction by which changes in protein function due to air pollution propagate through cellular pathways, complementing the traditional state of the art on air pollution-related disease to gain a more holistic view of molecular information on diseases caused by air pollution [12].

Recent progress in bioinformatics and computational biology has now made it possible to analyze these huge datasets generated from chemical–protein interactions and PPI networks, thus making better predictions of the impact of air pollution on human health more feasible [13]. Predictive modeling techniques involved in machine learning algorithms and systems biology approaches enable researchers to combine data from various sources, such as clinical studies, environmental exposure assessments, and experiments in molecular biology [14]. Identifying the susceptible nodes and pathways in interaction networks can help predict possible health outcomes due to air pollution [15]. Additionally, they enable the identification of new therapeutic targets to lessen the adverse effects of air pollution and provide the pathway for public health interventions [16].

These various levels of molecular interactions between pollutants and biological systems serve as the basis for targeted strategies for mitigating the health effects of air pollution. The utilization of chemical–protein and protein–protein interaction networks enhance our understanding of not only the harmful effects of pollutants but also the development of much better prevention and treatment strategies [17]. With continued growth in the burden of air pollution, applying these advanced tools will be paramount to the protection of human health and the development of policies that will reduce exposure to air pollutants in the years to come. In such a way, researchers are in a better position to predict the possible outcomes of air pollution on human health and scientifically support one of the major environmental challenges facing humanity at present. Our study aimed to investigate the protein–protein interactions (PPIs) of different air pollutants within the human body and identify the impacts of air pollution on human health. This study developed computer models of chemical–protein interactions to facilitate the determination of biomarkers for the early detection of pollution-related disorders.

## 2. Methods

### 2.1. Network Retrieval

The interaction of chemical compounds commonly present in air pollution (CO_2_, CO, H_2_S, SO_2_, O_3_, and NO_x_) with human proteins was investigated using the STITCH 4.0 (stitch.embl.de/ (accessed on 10 May 2024)) web server [18]. This program constructs a cohesive network derived from the origins of protein–chemical interactions utilizing experimental data, pathway databases, drug-target databases, text mining, and drug-target prediction. The list of proteins interacting with those chemicals was subsequently analyzed to ascertain the protein–protein interactions (PPIs) within the same species. PPI was identified using the STRING 9.1 (string-db.org/ (accessed on 12 May 2024)) web server [19], which retrieves all known and anticipated protein interactions based on direct (physical) and indirect (functional) relationships.

### 2.2. Sequence Retrieval

The sequences of proteins identified by STITCH and STRING were retrieved in FASTA format from the NCBI protein database (http://www.ncbi.nlm.nih.gov/protein (accessed on 3 June 2024)). Repetition was avoided throughout the sequence retrieval.

### 2.3. Functional Annotation and Pathway Analysis

OmicsBox software v.3.0 was used to perform functional annotation and pathway (Reactome and KEGG) analysis of these sequences, following Mia et al. [19]. Sequences were evaluated for their association with biological processes, molecular function, and cellular components. This tool analyzed and annotated protein sequences depending on their functional properties and respective pathways. We selected CO and H_2_S for further examination due to their classification as acute toxins, distinguishing them from other substances [20].

### 2.4. Identification of Critical Proteins and Their Key Pathways

Concentrating on high-degree nodes, we employed degree centrality to determine the crucial proteins that interacted with CO and H_2_S that were collected from the STRING database, following Mistry et al. [21]. Degree centrality denotes the number of particular connections with the protein (node) displays. Highly central nodes, regarded as hubs, play a vital role in numerous biological processes [22]. The clusters and key interaction pathways of the most critical proteins were analyzed using the Enrichr server (maayanlab.cloud/Enrichr/, (accessed on 14 February 2025)).

### 2.5. Prediction of the Hypothetical Direct Interactions Between the CO and H_2_S and the Proteins Identified by STITCH

#### 2.5.1. Collection and Preparation of Protein Structures and Ligands

We retrieved 10 protein structures for CO and 4 for H_2_S from the AlphaFold protein structure database (alphafold.com/, (accessed on 14 February 2025)). Then, we collected ligand structures of CO and H_2_S from the PubChem database and converted the ligand format (SDF to PDBQT) by the PyMol visualizer v.3.1.3.1. The ligand geometries were further optimized for structure and energy using Avogadro tools (Version 1.2.0) [23]. Then, the different properties, including physiochemical, lipophilicity, pharmacokinetics, and water solubility, were analyzed by SwissADME (http://www.swissadme.ch/index.php (accessed on 14 February 2025)). Further, the SCFBio server (Supercomputing Facility for Bioinformatics and Computational Biology, IIT Delhi, New Delhi, India) predicted the Lipinski Rule of 5. Additionally, these protein structures and ligands were prepared using MGL tools (Python Molecule Viewer) v.1.5.7. 

#### 2.5.2. Molecular Docking Studies

Molecular docking was performed using AutoDock Vina v.1.1.2 and visualized with BIOVIA Discovery Studio Visualizer v.24.1.0 [24].

## 3. Results

### 3.1. Identification of the Impactful Airborne Chemicals

This study investigated the interactions of carbon monoxide (CO), hydrogen sulfide (H_2_S), volatile carbon (VOC), sulfur dioxide (SO_2_), ozone (O_3_), and nitrogen oxides (NO_X_) with several proteins within the human body using the STITCH 4.0 and STRING 9.0 tools for protein–protein interactions (PPIs). Then, we continued this study with two airborne chemicals (CO and H_2_S) because these had the most beneficial and adverse effects on human health (Table 1 and Table 2) among all selected chemicals. The health impacts of other compounds are presented in the Appendix A.

### 3.2. Identification of Human Proteins That Interacted with Carbon Monoxide

Protein–chemical interaction (PCI) analysis revealed ten proteins identified by STITCH: ADI1, BLVRB, CYGB, HBB, HBE1, HBG1, HBG2, HMOX1, HMOX2, and SLC46A1. Additionally, these proteins interacted with an additional 65 proteins identified by STRING, as shown in Figure 1. The identified proteins are involved in regulating crucial biological functions such as methionine synthesis (ADI1), heme metabolism (BLVRB), cellular protection (CYGB), etc. (Table 1). Notably, the interacted proteins involved in heme catabolism and oxidative stress defense (HMOX1 and HMOX2). At low concentrations (< 10 ppm), CO positively influences human health, including inducing erythropoiesis, hemoglobin mass, and methionine metabolism. However, at higher concentrations (> 10 ppm), CO negatively affects cytochrome c oxidase activity, alters cellular protection, impairs heme metabolism, etc. (Table 1). This damage can lead to serious health issues, including respiratory problems, brain damage, and impaired cognitive function, highlighting the importance of reducing CO emissions for public health (Figure 2).

#### Functional Annotation and Pathway Analysis of CO-Interacted Proteins

Functional annotation demonstrated that interacted proteins are involved in the various biological processes such as O_2_ transport, transmembrane transport, signaling, immune system process, and stress response (Figure 3A). This aligns with previous reports on the influence of CO on molecular functions like oxidoreductase activity, hydrolase activity, etc. (Figure 3B). Additionally, these proteins interact with various cellular components, including cytosol, extracellular space, and mitochondrion (Figure 3C). Furthermore, this compound is associated with numerous Reactome pathways, including iron uptake and transport, post-translational protein phosphorylation, synthesis of GPI-anchored proteins, platelet degranulation, and heme biosynthesis, listed in Table 3. 

### 3.3. Identification of Human Proteins That Interacted with Hydrogen Sulfide

Five important proteins that interacted with H_2_S were found by PCI analysis: BHMT, CBS, CTH, MPST-3, and MTR-5. These proteins interacted with an additional 27 human proteins, identified from the STRING database through PPIs (Figure 4). These proteins play crucial roles in the regulation of human biological systems, including homocysteine metabolism (BHMT), methylation, DNA synthesis, folate metabolism (MTR-5), sulfur-containing compounds metabolism (MPST-3), etc. (Table 2). At lower concentrations (<100 µM), H_2_S positively impacts human health by influencing homocysteine metabolism and controlling the methionine cycle (Table 2). However, at elevated concentrations (>100 µM), it inhibits transcription and induces oxidation of these enzymes. Excessive hydrogen sulfide (H_2_S) can have detrimental effects on human health by causing damage to vital organs (Figure 5).

#### Functional Annotation and Pathway Analysis of H_2_S-Interacted Proteins

It has been shown that the functional annotation of H_2_S-interacted proteins is linked with numerous biological processes, cellular components, and molecular functions, including signaling, the metabolism of amino acids, programmed cell death, mitochondria, cytosol, sulfur transfer, and catalytic activity (Figure 6). Furthermore, this compound is associated with numerous Reactome pathways, including gluconeogenesis, the methionine salvage pathway, choline synthesis, and the methylation process, listed in Table 4.

### 3.4. Critical Human Proteins That Interacted with CO and H_2_S

Table 5 shows interactions involving CO, a total of 65 proteins that were identified through STRING analysis, with 15 exhibiting the highest degree of centrality. It is important to highlight that FECH, HMOX1, and ALB emerged as significant proteins due to their remarkable degree of centrality, and they play essential roles in heme metabolism, transportation, and cellular immunity. Furthermore, in the H_2_S interactions, a total of 27 proteins were identified from the STRING database, with 9 showing the highest degree of centrality (Table 5). Notably, CTH, CBS, and CBSL were identified as key proteins based on their extreme degree of centrality, and they play essential roles in serine, cysteine, and sulfur-containing amino acid metabolism, transportation, and cellular immunity. The analysis of the critical proteins using the cytoHubba plugin of Cytoscape revealed that the most significant proteins (FECH, HMOX1, ALB, CTH, CBS, and CBSL) exhibited a deep red color, as illustrated in Figure 7.

### 3.5. Key Pathways That Are Regulated by Identified Critical Human Proteins

The Reactome clustergram of FECH, HMOX1, and ALB indicates that these proteins predominantly influence the 10 essential Reactome pathways (Figure 8A). However, the *p*-value analysis reveals a strong interaction of these proteins with porphyrin metabolism (Table 6). Alternately, the KEGG clustergram showed these proteins interacted with nine KEGG pathways (Figure 8B), but based on the *p*-value, it was shown that these proteins strongly regulate the metabolism of the porphyrin and chlorophyll pathways (Table 6).

Alternately, the clustergram of the Reactome pathways of CTS and CBS showed that these proteins primarily regulate six crucial Reactome pathways (Figure 9A), but based on the *p*-value, it was shown that these proteins strongly interacted with the metabolism of ingested SeMet, Sec, and MeSec into H_2_Se (Table 7). Alternately, the KEGG clustergram showed these proteins interacted with three KEGG pathways (Figure 9B), but based on the *p*-value, it was shown that these proteins strongly regulated the glycine, serine, and threonine metabolism pathways (Table 7).

### 3.6. Hypothetical Direct Interactions Between CO and H_2_S and the Proteins Identified by STITCH

#### 3.6.1. Collection and Preparations of Protein Structures and Ligands

The protein structures (PDBs) of ADI1, BLVRB, CYGB, HBB, HBE1, HBG1, HBG2, HMOX1, HMOX2, and SLC46A1 that interacted with CO and those of BHMT, CBS, CTH, MPST-3, and MTR-5 for H_2_S was collected from the AlphaFold Protein Structure Database, and then prepared and visualized by MGL tools (AutoDock visualizer) v.1.5.7.

#### 3.6.2. SwissADME Results for CO and H_2_S Airborne Chemicals

The SwissADME identified the properties of two ligands (chemicals): 0.00Å^2^ TPSA (Topological Polar Surface Area) for CO and 25.30 Å^2^ for H_2_S; they have no rotatable bonds and have lead likenesses with poor GI absorption (Table 8). The results of Lipinski’s Rule of 5 for the CO and H_2_S ligands are shown in Table 9.

#### 3.6.3. Results of Hypothetical Protein–Chemical Interactions

AutoDock Vina was used to dock the 2 ligands (CO and H_2_S) with the 14 targeted proteins. The docking results (binding energy/affinity) are demonstrated in Table 10. Among the proteins interacting with CO, ADI1 exhibited the highest binding energy (−1.9 kcal/mol). CO mainly interacted with glycine, isoleucine, histidine, and aspartate residues by the convention hydrogen bond and carbon–hydrogen bond. All the docking results are illustrated in Figure 10, but these interacting energies are inadequate for robust docking. Comparable findings were noted in the docking results of proteins interacting with H_2_S, and their binding energies are very low (−0.8 kcal/mol is the highest binding affinity); therefore, it was not visualized in Discover Studio.

## 4. Discussion

Carbon monoxide (CO) and hydrogen sulfide (H_2_S) are important gaseous molecules with significant positive and negative biological effects. Even though they are known to be poisonous in greater quantities (over 4 mg/m^3^ or 3.5 ppm), they are also essential for preserving cellular homeostasis and controlling several physiological functions [45]. With an emphasis on their interactions with important proteins, oxidative stress management, and consequences for vascular and metabolic health, this discussion will examine how CO and H₂S affect enzyme activity, cellular pathways, and metabolic activities.

The major effects of CO on biological systems are through the interaction with enzymes, proteins, and cellular pathways responsible for a variety of key physiological functions. Among the well-defined targets of CO are protein arginine methyltransferases, which catalyze protein methylation and control many cellular processes, from gene expression to cell signaling and cell cycle control. 

Other important effects of CO involve interactions with proteins implicated in the regulation of oxidative stress and reactive oxygen species. CO modulates the activity of enzymes that are central in the oxidative balance, including HO-1 and HO-2, which are crucial for heme catabolism and the production of biliverdin and carbon monoxide itself. Of these, HO-1 possesses antioxidant properties and is induced by conditions of oxidative stress. CO, through the modulation of HO-1, may have protective effects against oxidative injury [46]. On the other hand, chronic CO exposure could interfere with cellular homeostasis and promote oxidative injury in conditions such as neurodegenerative disorders, mainly by breaking down the integrity of the blood–brain barrier [47]. Such breakdown in the integrity of the blood-brain barrier (BBB) may lead to enhanced neuroinflammatory responses and is considered an additional factor contributing to cognitive decline.

Further, CO is a player in the processes of vascular homeostasis. It modifies the functions of important oxygen-transport proteins, including cytoglobin and hemoglobin, implicated in oxygen transport and the maintenance of tissue oxygen levels. With the ability to modulate these proteins, CO modifies the delivery of oxygen to the tissues and the cellular responses to hypoxia; as a consequence, the blood flow is changed, as are tissue repair and immune responses [48]. Moreover, CO acts as an essential modulator of immune responses by tuning the production of inflammatory mediators with protective or pathogenic functions, depending on the context and length of exposure. While in an acute setting, CO can reduce inflammation, on the other hand, chronic exposure may increase inflammatory pathways that may lead to the development of chronic diseases like cardiovascular disorders and neurodegenerative conditions.

This study discovered that carbon monoxide interacts with multiple human proteins, genes, and enzymes that regulate vital functions of human health within tolerable concentrations, including methionine synthesis (ADI1), heme metabolism (BLVRB), and cellular protection (CYGB). Carbon monoxide is produced during heme metabolism by BLVRB enzymes, and extensive amounts of this airborne chemical alter cellular respiration [26]. Acireductone dioxygenase 1 (ADI1) is responsible for the methionine salvage pathway crucial for methionine synthesis and cellular stress responses [25]. Low CO levels (10–100 nM) influence methionine metabolism, but high CO levels (>100 µM) can alter enzyme metal centers, such as the iron (Fe^2^⁺) in ADI1, reducing catalytic activity [25]. Another primary interacted protein, cytoglobin (CYGB) protein, is responsible for cellular protection via antioxidant defense, lowering blood pressure, maintaining vascular health via nitric oxide regulation, and promoting cardiac cell survival [49]. Hemoglobin subunit epsilon 1 (HBE1) plays a crucial role in treating different hematologic disorders and blood cancer, but extensive CO negatively affects the functions of HBE1. Hemoglobin subunit gamma 1 (HBG1) plays a crucial role in fetal oxygen transport, but an extensive concentration of CO (>10 ppm) increases hemoglobin’s affinity for oxygen and causes impaired oxygen delivery to tissues. The enzyme betaine homocysteine S-methyltransferase (BHMT) interacts with H_2_S and is involved in the conversion of homocysteine into methionine via betaine methyl donor [36].

Like CO, hydrogen sulfide has also been recognized as an important gaseous molecule in biological systems with significant roles, primarily in regulating metabolic and redox processes. The role of H₂S is being reported to influence a lot of enzymes in sulfur metabolism, most especially in the trans-sulfuration pathway, which has an important role in the detoxification of homocysteine and in synthesizing major sulfur-containing biomolecules. Two important enzymes from the trans-sulfuration pathway are BHMT and CBS. Both of these enzymes are essential to converting homocysteine to cysteine and further synthesizing glutathione, an important antioxidant [50]. H_2_S, through its action on these enzymes, maintains the redox balance and thus supports the maintenance of cellular metabolism, reducing oxidative damage due to ROS. H_2_S has a beneficial impact on homocysteine metabolism through the induction of trans-sulfuration enzymes [37]. The enzyme cystathionine gamma-lyase (CTH) plays an essential role in homocysteine metabolism. A lower concentration (<10 µM) of H_2_S increases CTH function by inducing the feedback inhibition mechanism, but a higher concentration (>10 µM) of H_2_S decreases transcription and reduces the activity of the CTH enzyme (200 µM) [40]. MPST-3 plays a key role in the sulfur-containing compounds, and Nagahara [42] said the elevated levels (200 µM) of H_2_S induce oxidation of the MPST-3 enzyme and minimize its activity.

The possibility that H₂S can regulate oxidative stress is intriguing. Since it is a reducing molecule, H₂S can lessen oxidative damage by either directly increasing the cell’s antioxidant defense systems or by scavenging reactive oxygen species (ROS). For example, H₂S increases the expression and activity of glutathione peroxidase and other antioxidant enzymes, thus protecting the cell from oxidative damage [51]. This antioxidative role is highly valued in the context of diseases associated with inflammation and oxidative damage, such as cardiovascular diseases, diabetes, and neurodegenerative disorders.

Furthermore, H_2_S plays a role in energy production and metabolic efficiency. By influencing the activity of several enzymes like MAT and methionine synthase, which are responsible for synthesizing important metabolic intermediates, H_2_S provides a means to maintain energy homeostasis in the cell. Thus, a further suggestion was made about its ability to ensure an optimum metabolic outcome and to improve mitochondrial process efficiency [52]. These would have implications for CBS and cystathionine that further support the synthesis of sulfur and methionine, which are important in cellular growth, repair, and energy production. The beneficial roles played by H₂S in the regulation of oxidative stress and inflammation point toward its therapeutic potential in various conditions, particularly where such processes are dysregulated.

FECH, ALB, and HMOX1 are essential proteins for human health that play a significant role in regulating porphyrin and heme metabolism. These proteins have been identified as critical nodes based on their degree of centrality. Finding the most important nodes in a network based on the degree of centrality allows one to examine its topology, including its resilience and susceptibility to attacks [53]. Centrality measures employ graph theory and network analysis to evaluate an individual’s position within a protein network. Degree centrality, betweenness centrality, and closeness centrality are used to assess social influence inside networks [54]. Molecular docking interactions indicated the lowest binding energy, suggesting that CO and H_2_S have a weak interaction with these 14 proteins. Multiple investigations have explored the strong binding of CO with the heme pocket of myoglobin; however, we were unable to locate myoglobin within the STITCH database [55]. CO mostly binds with glycine and aspartate amino acids, and these are responsible for erythropoiesis and oxidative responses. Alternatively, a limited investigation was performed on the interactions of H_2_S with proteins; Nery et al. [56] reported comparable results regarding H_2_S interactions with hypoxia-inducible factors.

Both CO and H₂S are therapeutically very important. CO’s role in modulating vascular health, immune responses, and oxidative stress could represent targeted therapies for conditions like hypertension, vascular diseases, and neurodegenerative disorders. CO-releasing molecules, CORMs, have been pursued as active pharmaceuticals for the delivery of controlled amounts of CO for therapeutic applications, especially in diseases that feature excessive oxidative stress and inflammation.

However, although both gases possess therapeutic potential, dosing and delivery mechanisms are still big challenges. At high concentrations, both CO and H₂S are toxic; thus, it is very hard to translate their beneficial effects into clinical practice. For this reason, ongoing research is needed to develop ways to safely and effectively exploit these molecules in therapeutic settings. However, the therapeutic potential of these gases must be carefully harnessed to avoid their toxic effects, underscoring the importance of continued research in this area.

## 5. Conclusions

The increases in greenhouse gas emissions due to urbanization and industrial activities have been linked to a significant fall in air quality, now regarded as a foremost risk to public health globally. The emission of pollutants including particulate matter (PM), nitrogen dioxide (NO_2_), sulfur dioxide (SO_2_), carbon monoxide (CO), hydrogen sulfide, and volatile organic compounds (VOCs) has worsened air quality and is implicated in diverse health effects, predominantly on the respiratory and cardiovascular systems. These pollutants induce climate change and release a series of cellular and molecular effects in the human body, which lead to chronic diseases like asthma, chronic obstructive pulmonary disease (COPD), and heart disease. The priority in environmental health research has been to better understand these health effects at the molecular level. In particular, studies concentrating on the interactions of these pollutants with proteins at the molecular level will illuminate how these compounds throw biological functions out of gear and lead to the development of such diseases.

Air pollutants harm and benefit human health by interacting with cellular proteins. These cellular proteins are critical in maintaining normal physiological functions. Once inhaled, air pollutants bind to proteins, modifying their conformation, activity, or expression. In turn, this process disrupts essential biological functions such as immune responses, oxidative stress regulation, and cell signaling. Oxidative stress, a prominent mechanism through which air pollutants cause damage, is mediated largely by proteins involved in antioxidant defense mechanisms. Chemistry–protein interaction analysis is a very innovative tool to unveil the biological effects of air pollution. Scientists can get insight into the molecular pathways that are implicated after exposure by identifying the specific proteins that interact with different pollutants. In recent years, advances in bioinformatics and computational biology have allowed researchers to analyze the vast data associated with these chemical–protein interactions.

The research discusses the interaction between carbon monoxide (CO) and various proteins subsequently modifying several biological functions like methionine biosynthesis, heme metabolism, and cell protection; for instance, CO interacts with acireductone dioxygenase 1 (ADI1) and biliverdin reductase B (BLVRB), which has various negative effects on several cellular and metabolic functions. CO at low concentrations (about 10–15 ppm) seems to enhance erythropoiesis and affects methionine metabolism, while higher concentrations (>50 ppm) interfere with the proper functioning of hemoglobin, meaning cellular respiration is affected, and oxidative stresses occur. Such interactions may bring serious implications for health, such as respiratory and cognitive defects. In addition, the involved proteins may link with different biological pathways such as heme biosynthesis, platelet degranulation, and oxidative stress defense, thus emphasizing that CO exposure can have a very far-reaching effect.

Hydrogen sulfide (H_2_S) also inhibits key enzymes involved in the metabolism of homocysteine and sulfur-containing compounds and methylation processes. H_2_S inhibits essential enzymes such as betaine homocysteine S-methyltransferase (BHMT) and cystathionine beta-synthase (CBS), which disrupts proper metabolism. A low dose of H_2_S stimulates homocysteine metabolism and the methionine cycle, while a higher concentration gradually oxidizes enzymes, inhibits transcription, and damages further cellular function and metabolism. The critical pathways’ interaction with H_2_S mainly takes part in sulfur metabolism, methylation, and the regulation of homocysteine.

Also identified were critical proteins with high centralities in the network of protein interactions for both CO and H_2_S. For CO, involved in heme metabolism, cellular immunity, and stress response are proteins such as FECH, HMOX1, and ALB, while for H_2_S, CTH, CBS, and CBSL are involved in sulfur metabolism, amino acid regulation, and redox balance. The study further elaborates on these proteins’ regulatory roles in diverse biochemical processes, such as porphyrin metabolism and sulfur amino acid metabolism, as well as the regulation of gene expression, all of which emphasize how CO and H_2_S exert profound effects on human well-being.

In conclusion, molecular interactions between pollutants and proteins, in terms of their biological effects, are undoubtedly of great importance for understanding the health consequences of air pollution. Large strides forward in bioinformatics and computational biology, including the STITCH 4.0 and STRING 9.0 tools, have made it possible to elucidate the chemical–protein interaction networks and protein–protein interaction pathways and elaborate the molecular mechanisms of air-pollution-related diseases. Further digging into these networks and the disruption caused to biological processes may help to determine health outcomes, suggest new therapeutic approaches, and possibly prevent or ameliorate air pollution’s harmful effects. This type of research is certainly of the utmost importance in matters of public health and will form a stronger basis for understanding how the environment can influence human disease.

## 6. Future Direction/Recommendations

Moreover, the study shows the urgent need for further research into the health effects of air pollution at the molecular level. As air pollutants interact with human proteins, future research needs to be oriented toward a deeper understanding of the biochemical networks. Guided by the results arising from this study, several recommendations are possible:Expand the study of molecular interaction: While this study used STITCH 4.0 and STRING 9.0 to investigate the interaction of proteins with H_2_S and CO pollutants, future studies need to be expanded in scope to other common pollutants such as particulate matter, nitrogen dioxide, and sulfur dioxide. These are common in many industrial and urban areas and affect protein function, leading to a variety of diseases. Advanced computational models can be utilized in simulating their interaction with proteins, hence enabling the researchers to find more specific pathways affected by these pollutants.Investigation into long-term exposure effects: Most studies generally focus on acute exposure to such pollutants, but their long-term exposures have significant cumulative health effects. It is crucial to determine how chronic exposure to these pollutants alters protein functions over time and how such alterations may lead to the onset of various chronic diseases such as asthma, cardiovascular diseases, and even cancer. Longitudinal studies following the molecular changes in exposed individuals can offer a better understanding of such long-term effects.Multidisciplinary engagement: The nature of the challenge in studying protein–protein interaction networks require merging expertise from bioinformatics, molecular biology, environmental science, and epidemiology in designing studies that connect changes at the molecular level to population-level measurable health outcomes; this could be further extended in collaborative work toward improving the predictive ability of models for forecasting health risk from environmental pollution data.Public health and policy implications: The understanding of molecular interactions between pollutants and proteins can help in the development of targeted therapies. Governments and public health organizations should prioritize research that links environmental pollution to specific health conditions and implement policies that reduce exposure. Such policy measures may include strict regulation of emissions, increased monitoring of air quality, and public health campaigns to raise community awareness about the risks of pollution.Development of therapeutic interventions: The identification of key proteins affected by pollutants will provide the window for developing therapeutic interventions. Drug development should be based on mitigating the harmful effects of pollutants on the proteins. Further research should be conducted to develop chemicals or interventions that can protect critical proteins from pollution-induced damage and assist in reducing the chance of developing pollution-related disorders.

Addressing the health effects of air pollution necessitates a multitherapy approach: detailed molecular research, public health campaigns, and policy alteration. Understanding the interaction of pollutants with proteins opens new avenues to prevent or treat diseases associated with air pollution. 

## Figures and Tables

**Figure 1 ijerph-22-00418-f001:**
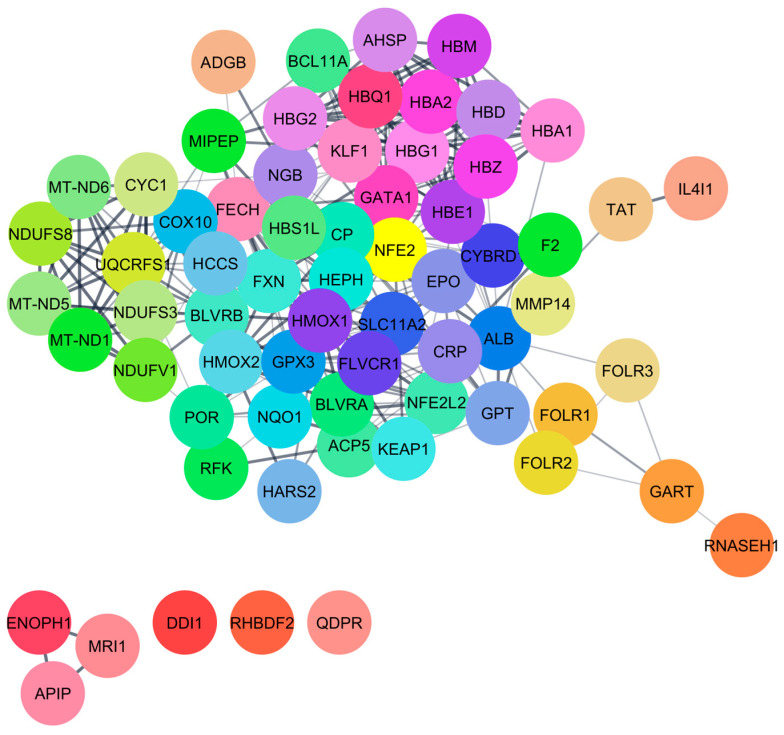
Interacted proteins of CO identified from the STRING database.

**Figure 2 ijerph-22-00418-f002:**
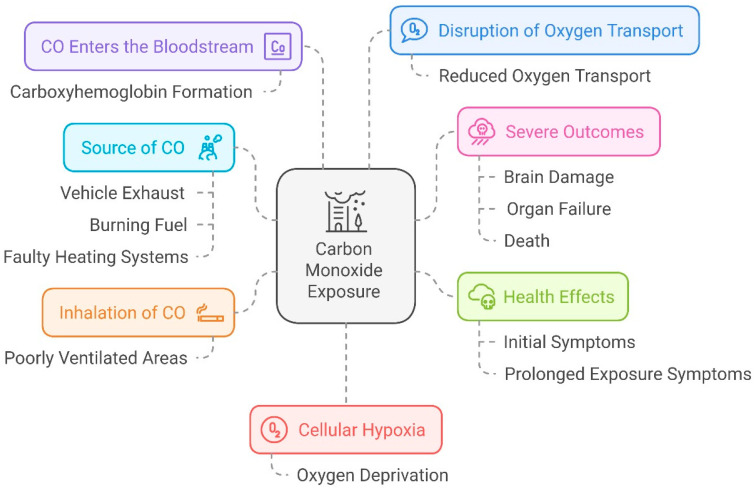
CO affects several organs and their functions.

**Figure 3 ijerph-22-00418-f003:**
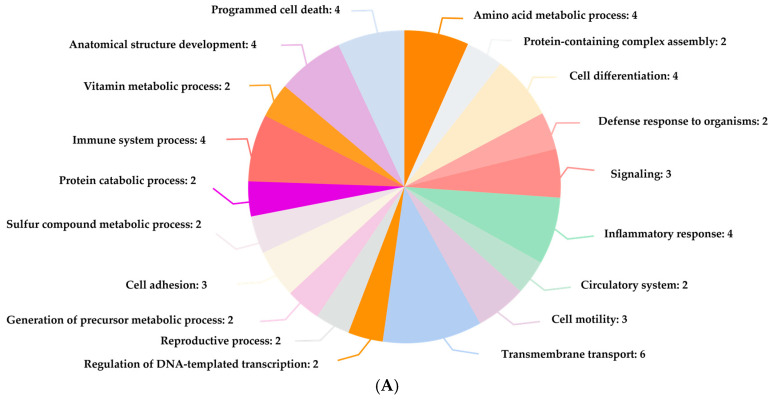
Functional annotation study revealed CO-interacted proteins involved in various (**A**) biological processes, (**B**) molecular functions of distinct proteins, and (**C**) cellular component localization in *Homo sapiens*.

**Figure 4 ijerph-22-00418-f004:**
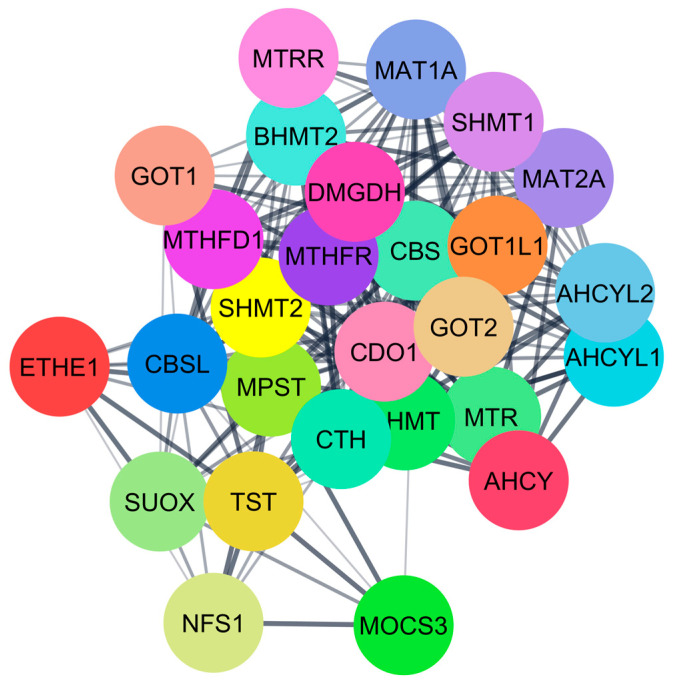
Proteins that interacted with H_2_S were identified by STRING.

**Figure 5 ijerph-22-00418-f005:**
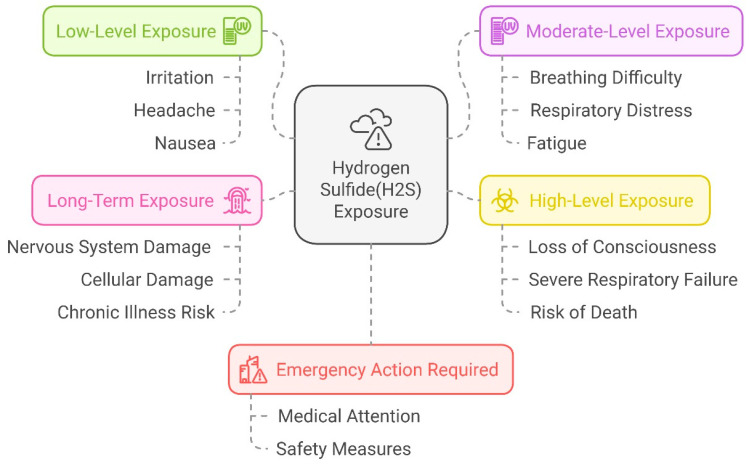
Hydrogen sulfide (H_2_S) affects several human organs.

**Figure 6 ijerph-22-00418-f006:**
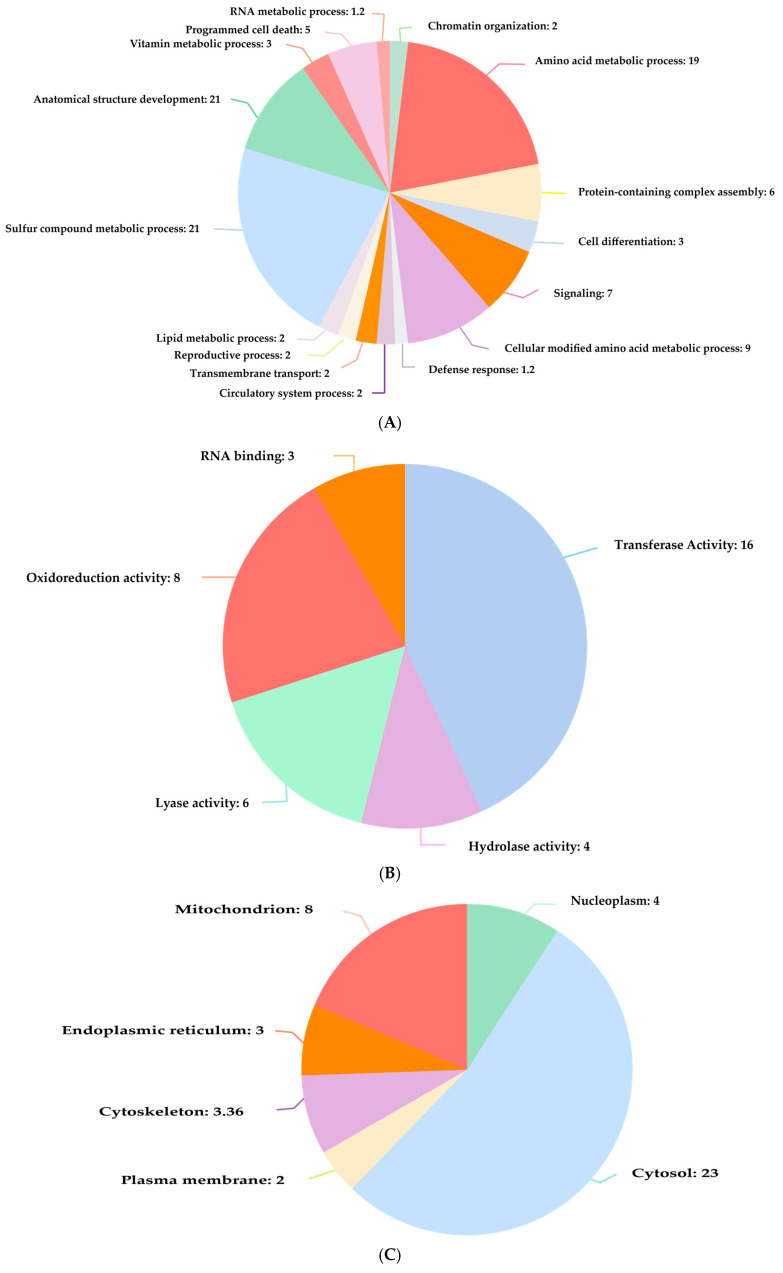
Functional annotation study revealed H_2_S-interacted proteins involved in various biological processes (**A**), molecular functions (**B**), and cellular component localization (**C**).

**Figure 7 ijerph-22-00418-f007:**
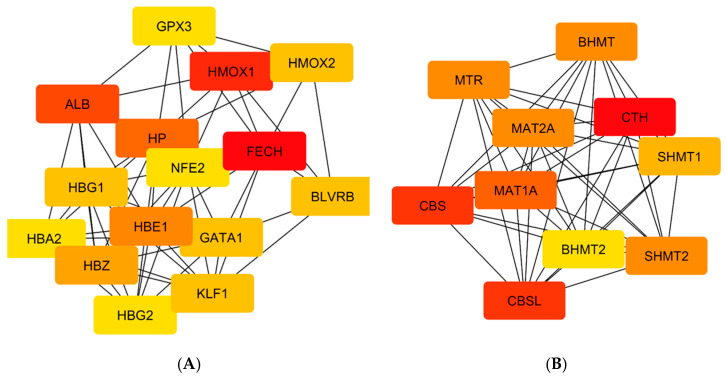
Critical nodes of human proteins that interacted with CO (**A**) and H_2_S (**B**); deep red color indicating the most crucial nodes (FECH, HMOX1, ALB, CBS, CBSL, and CTH).

**Figure 8 ijerph-22-00418-f008:**
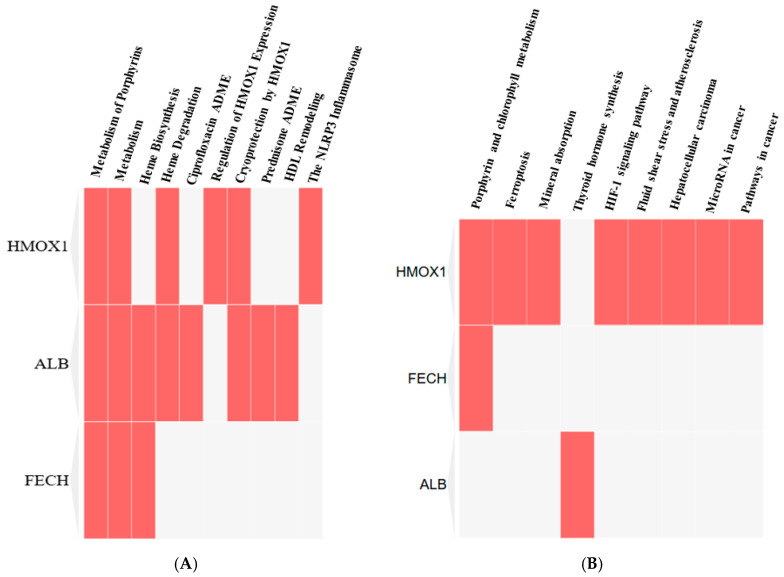
Clustergram analysis of Reactome (**A**) and KEGG (**B**) pathways regulated by FECH, HMOX1, and ALB.

**Figure 9 ijerph-22-00418-f009:**
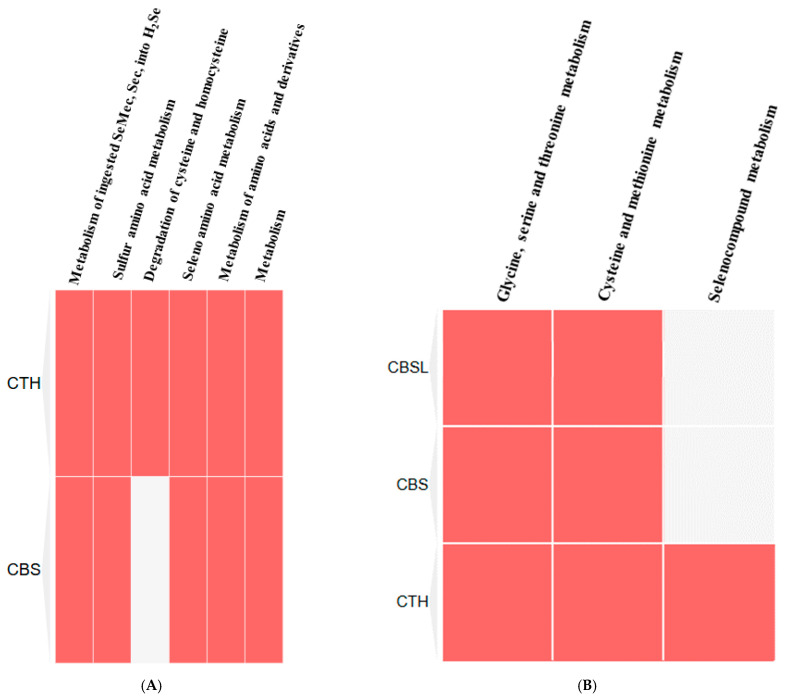
Clustergram analysis of Reactome (**A**) and KEGG (**B**) pathways regulated by CTH, CBS, and CBSL.

**Figure 10 ijerph-22-00418-f010:**
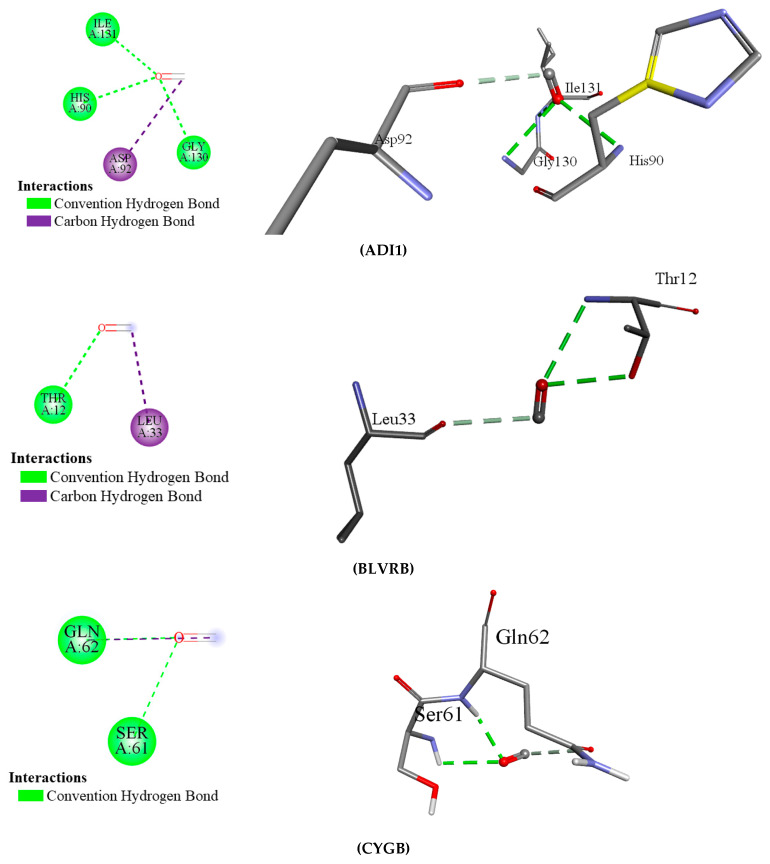
(**A**) 2D interaction of CO with target proteins. (**B**) 3D interaction of CO with target proteins using AutoDock Vina and visualized by Discovery Studio.

**Table 1 ijerph-22-00418-t001:** Comprehensive analysis of carbon monoxide (CO) impacts on various aspects of biological systems, including enzymes, pathways, proteins, and body functions.

Protein	Function	Positive Effects of CO on Protein Functions	Negative Effects of CO on Protein Functions
Acireductone dioxygenase 1 (ADI1)	Responsible for the methionine salvage pathway, which is crucial for methionine synthesis and cellular stress responses.	Low CO levels (1–20 ppm) influence methionine metabolism and associated pathways, affecting cellular homeostasis and ADI1.	High CO levels (>20 ppm) can alter enzyme metal centers, such as the iron (Fe^2^⁺) in ADI1, reducing catalytic activity. It may impair cellular metabolism by altering the methionine salvage pathway [25].
Biliverdin reductase B (BLVRB)	Responsible for heme metabolism, antioxidant defense, and cellular signaling.	Unknown	By attaching to its heme-iron core, CO can reduce cytochrome c oxidase, therefore interfering with cellular respiration and perhaps influencing cellular metabolic processes by blocking BLVRB function [26].
Cytoglobin (CYGB) protein	Responsible for cellular protection via antioxidant defense, maintaining vascular health via nitric oxide regulation and promoting cardiac cell survival [27].	Unknown	Unknown
Hemoglobin subunit beta (HBB)	It is an essential component of hemoglobin and is present in red blood cells; it transports oxygen throughout the body.	Lower concentration (approximately 5%) of CO positively affects HBB protein and enhances erythropoiesis and hemoglobin mass [28].	Higher concentration of CO increases hemoglobin’s affinity for oxygen and causes impaired oxygen delivery to tissues.
Hemoglobin subunit epsilon 1 (HBE1)	Plays a crucial role in embryonic development and is used in treating different hematologic disorders and blood cancers [29].	Unknown	CO negatively affects the functions of the HBE1 protein by binding with the hemoglobin subunit and causing hypoxia.
Hemoglobin subunit gamma 1 (HBG1)	Plays a crucial role in fetal oxygen transport [30].	Unknown	Higher concentration of CO (50–150 ppm) increases hemoglobin’s affinity for oxygen and causes impaired oxygen delivery to tissues.
Hemoglobin subunit gamma 2 (HBG2)	Plays a crucial role in fetal oxygen transport [30].	Unknown	Higher concentration (50–150 ppm) of CO increases hemoglobin’s affinity for oxygen and causes impaired oxygen delivery to tissues by affecting HBG2 functions.
The enzyme heme oxygenase 1 (HMOX1)	Essential for biliverdin production from heme breakdown and oxidative stress protection [31].	Lower CO (10–15 ppm) concentration regulates immunological responses by inhibiting the proliferation of T cells and the generation of interleukin-2 [32].	Higher CO (>50 ppm) concentration can reduce HMOX1 activity and disturb its positive roles, causing possible toxicity and more oxidative stress from the heme-accumulating effect [33].
Heme oxygenase 2 (HMOX2)	Essential for biliverdin, free ion, and CO production from heme breakdown [34].	Lower (10–15 ppm) concentration of CO regulates immunological responses by inhibiting the proliferation of T cells and the generation of interleukin-2 [32].	Higher CO (>50 ppm) concentration can reduce HMOX2 activity and disturb its positive roles [33].
Solute carrier family 46 member 1 (SLC46A1)	Plays a crucial role in the transport and metabolism of folate and other molecules across cell membranes (intestines and central nervous system) [35].	Unknown	Higher concentration of CO (>150 ppm) affects SLC46A1 activity by altering the proton-coupled folate transporter (PCFT).

**Table 2 ijerph-22-00418-t002:** Comprehensive analysis of the impacts of hydrogen sulfide (H_2_S) on various aspects of biological systems, including enzymes, pathways, proteins, and body functions.

Protein	Function	Positive Effects of H_2_S on Protein Functions	Negative Effects of H_2_S on Protein Functions
The enzyme betaine homocysteine S-methyltransferase (BHMT)	Involved in the conversion of homocysteine into methionine via betaine methyl donor [36].	H_2_S has a beneficial impact on homocysteine metabolism through the induction of trans-sulfuration enzymes [37].	Unknown.
Cystathionine beta-synthase (CBS)	Plays a crucial role in homocysteine metabolism, redox balance, and H_2_S production [38].	Unknown.	Unknown.
The enzyme cystathionine gamma-lyase (CTH)	Plays an essential role in homocysteine metabolism with the production of cysteine, ammonia, and α-ketobutyrate [39].	A lower concentration (<10 µM) of H_2_S increases CTH function by inducing the feedback inhibition mechanism [40].	Higher concentrations (>10 µM) of H_2_S decrease the transcription and activity of the CTH enzyme (200 µM).
3-Mercaptopyruvate sulfurtransferase (MPST-3)	Plays a key role in the metabolism of sulfur-containing compounds (Thiols, sulfides, etc.) and H_2_S production [41].	The minimum amount (<100 µM) of H_2_S plays a crucial role in influencing the activity of the MPST-3 enzyme [42].	Elevated levels (200 µM) of H_2_S induce oxidation of the MPST-3 enzyme and minimize its activity [42].
Methionine synthase reductase isoform 5 (MTR-5)	It is a key enzyme of homocysteine to methionine conversion, methylation, DNA synthesis, and folate metabolism [43].	The minimum amount of H_2_S (10 nM to 100 nM) positively influenced homocysteine catabolism and maintained the overall functions of the methionine cycle [44].	Unknown.

**Table 3 ijerph-22-00418-t003:** Pathways associated with the CO-interacted proteins.

Reactome ID	Pathway’s Name	Reactome ID	Pathway’s Name
R-HSA-917937	Iron uptake and transport	R-HSA-9818749	Regulation of NFE2L2 gene expression
R-HSA-1592389	Activation of Matrix Metalloproteinases	R-HSA-114608	Platelet degranulation
R-HSA-1247673	Erythrocytes take up oxygen and release carbon dioxide	R-HSA-611105	Respiratory electron transport
R-HSA-6807878	COPI-mediated anterograde transport	R-HSA-8964058	HDL remodeling
R-HSA-425410	Metal ion transporters	R-HSA-6785807	Interleukin-4 and interleukin-13 signaling
R-HSA-1442490	Collagen degradation	R-HSA-379726	Mitochondrial tRNA aminoacylation
R-HSA-9700645	ALK mutants bind TKIs	R-HSA-1268020	Mitochondrial protein import
R-HAS-163125	Post-translational modification of GPI-anchored proteins	R-HSA-9662834	CD163 mediating an anti-inflammatory response
R-HSA-1234158	Regulation of gene expression by hypoxia-inducible factor	R-HSA-2168880	Scavenging of heme from plasma
R-HSA-6799198	Complex I biogenesis	R-HSA-189451	Heme biosynthesis
R-HSA-5655799	Defective SLC40A1 causes hemochromatosis 4 (HFE4)	R-HSA-9679191	Potential therapeutics for SARS
R-HSA-9707587	Regulation of HMOXI1 expression and activity	R-HSA-8964208	Phenylalanine metabolism
R-HSA-3299685	Detoxification of reactive oxygen species	R-HSA-9617828	FOXO-mediated transcription of cell cycle genes
R-HSA-203615	eNOS activation	R-HSA-5689880	Ub-specific processing proteases
R-HSA-1362409	Mitochondrial iron–sulfur cluster biogenesis	R-HSA-9818027	NFE212 regulating detoxification enzymes
R-HSA-9707564	Cytoprotection by HMOX1	R-HSA-189483	Heme metabolism
R-HSA-167827	The proton buffering model	R-HSA-8957275	Post-translational protein phosphorylation
R-HSA-9755511	KEAP-1NFE2L2 pathway	R-HSA-379726	Mitochondrial tRNA aminoacylation
R-HSA-8981607	Intracellular oxygen transport	R-HSA-173623	Classical antibody-mediated complement activation
R-HSA-196757	Metabolism of folate and pteridines	R-HSA-196843	Vitamin B2 (riboflavin) metabolism
R-HSA-5694530	Cargo concentration in the ER	R-HSA-8951664	Neddylation
R-HSA-73817	Purine ribonucleoside monophosphate biosynthesis	R-HSA-159418	Recycling of bile acids and salts
R-HSA-1237044	Erythrocytes take up carbon dioxide and release oxygen	R-HSA-983231	Factors involved in platelet production
R-HSA-8980692	RHOA GTPase cycle	R-HSA-167827	The proton-buffering model
R-HSA-211897	Cytochrome P450—arranged by substrate type	R-HSA-9749641	Aspirin ADME
R-HSA-9757110	Prednisone ADME	R-HSA-429958	miRNA decay by 3’ to 5′ exoribonuclease
R-HSA-9793528	Ciprofloxacin ADME	R-HSA-173623	Classical antibody-mediated complement activation
R-HSA-6798695	Neutrophil degradation	R-HSA-9627069	Regulation of the apoptosome activity
R-HSA-1237112	Methionine salvage pathway	R-HSA-5619048	Defective SLC11A2 causes hypochromic microcytic anemia

**Table 4 ijerph-22-00418-t004:** Pathways associated with H_2_S-interacted proteins.

Reactome ID	Pathway	Reactome ID	Pathway
R-HSA-5579024	Defective MAT1A causes MATD	R-HSA-8963693	Aspartate and asparagine metabolism
R-HSA-1237112	Methionine salvage pathway	R-HSA-1614635	Sulfur amino acid metabolism
R-HSA-9013407	RHOH GTPase cycle	R-HSA-70263	Gluconeogenesis
R-HSA-9013408	RHOG GTPase cycle	R-HSA-2408508	Metabolism of ingested SeMet, Sec, MeSec into H_2_Se
R-HSA-196757	Metabolism of folate and pteridines	R-HSA-3359469	Defective MTR causes HMAG
R-HSA-156581	Methylation	R-HSA-71262	Carnitine synthesis
R-HSA-94758	Molybdenum cofactor biosynthesis	R-HSA-9759218	Cobalamin (Cbl) metabolism
R-HSA-8964539	Glutamate and glutamine metabolism	R-HSA-6798163	Choline catabolism
R-HSA-1614517	Sulfide oxidation to sulfate	R-HSA-425381	Bicarbonate transporters
R-HSA-1362409	Mitochondrial iron–sulfur cluster biogenesis	R-HSA-389661	Glyoxylate metabolism and glycine degradation

**Table 5 ijerph-22-00418-t005:** Critical human proteins with degree centrality that interacted with CO or H_2_S.

Carbon Monoxide			Hydrogen Sulfide		
Rank	Protein	Degree	Rank	Protein	Degree
1	FECH	26	1	CTH	25
2	HMOX1	25	2	CBS	24
3	ALB	24	3	CBSL	24
4	HP	21	4	MAT1A	20
5	HBE1	19	5	SHMT2	19
6	HBZ	18	6	MAT2A	19
7	GATA1	17	7	MTR	19
8	HBG1	17	8	BHMT	19
9	BLVRB	17	9	SHMT1	18
10	HMOX2	17			
11	KLF1	17			
12	HBA2	16			
13	NFE2	16			
14	HBG2	16			
15	GPX3	16			

**Table 6 ijerph-22-00418-t006:** Reactome and KEGG clustering pathways of FECH, HMOX1, and ALB that interacted with CO.

Rank	Reactome Pathway	*p*-Value	Related Protein
1	Metabolism of porphyrins	2.457 × 10^−9^	FECH, ALB, and HMOX1
2	Heme biosynthesis	0.000001364	FECH and ALB
3	Heme degradation	0.000001799	ALB and HMOX1
4	Cytoprotection by HMOX1	0.00002562	ALB and HMOX1
5	Cellular response to chemical stress	0.0003025	ALB and HMOX1
6	Ciprofloxacin ADME	0.0007498	ALB
7	Regulation of HMOX1 expression and activity	0.0007498	HMOX1
8	Metabolism	0.001295	FECH, ALB, and HMOX1
9	Prednisone ADME	0.001499	ALB
10	HDL remodeling	0.001499	ALB
KEGG pathway			
1	Porphyrin and chlorophyll metabolism	0.00001353	FECH and HMOX1
2	Ferroptosis	0.006138	HMOX1
3	Mineral absorption	0.008973	HMOX1
4	Thyroid hormone synthesis	0.01121	ALB
5	HIF-1 signaling pathway	0.01626	HMOX1
6	Fluid shear stress and atherosclerosis	0.02071	HMOX1
7	Hepatocellular carcinoma	0.02499	HMOX1
8	MicroRNAs in cancer	0.04579	HMOX1
9	Pathways in cancer	0.07756	HMOX1

**Table 7 ijerph-22-00418-t007:** Reactome and KEGG clustering pathways of CBS, CTH, and CSBL that interacted with H_2_S.

Rank	Pathway	*p*-Value	Related Protein
1	Metabolism of ingested SeMet, Sec, MeSec into H_2_Se	4.199 × 10^−7^	CBS and CTH
2	Sulfur-containing amino acid metabolism	0.000005665	CBS and CTH
3	Selenoamino acid metabolism	0.0001103	CBS and CTH
4	Metabolism of amino acids and derivatives	0.0009472	CBS and CTH
5	Metabolism of cysteine and homocysteine	0.002248	CTH
6	Metabolism	0.03307	CBS and CTH
KEGG pathway			
1	Glycine, serine, and threonine metabolism	7.410 × 10^−9^	CBS, CTH, and CSBL
2	Cysteine and methionine metabolism	1.470 × 10^−8^	CBS, CTH, and CSBL
3	Selenocompound metabolism	0.002548	CTH

**Table 8 ijerph-22-00418-t008:** Properties of ligands identified with SwissADME.

Ligand Name	Filters					
	Physicochemical Properties		Lipophilicity	Pharmacokinetics	Water Solubility	Lead Likeness
	TPSA (Å^2^)	No. of Rotatable Bonds	Consensus Log *p*	GI Absorption		
Carbon monoxide	0.00	0	−0.31	Low	Very soluble	No
Hydrogen sulfide	25.30	0	−0.28	Low	Soluble	No

**Table 9 ijerph-22-00418-t009:** Results of Lipinski’s Rule of 5 for the CO and H_2_S ligands.

Ligand Name	Property				
	Mass (mg/mol)	H Donor	H Acceptor	Log *p*	Molar Refractivity
Carbon monoxide	312.00	5	6	−0.053101	77.14
Hydrogen sulfide	34.00	0	0	0.11	10.38

**Table 10 ijerph-22-00418-t010:** AutoDock Vina Docking Results- mode and binding affinity for airborne chemicals (CO and H_2_S) with identified (by STITCH) 14 proteins.

Carbon Monoxide		
Protein	Mode	Binding Affinity (kcal/mol)
Acireductone dioxygenase 1 (ADI1)	1	−1.9
	2	−1.6
Biliverdin reductase B (BLVRB)	1	−1.8
	2	−1.7
Cytoglobin (CYGB)	1	−1.7
	2	−1.5
Hemoglobin subunit beta (HBB)	1	−1.6
	2	−1.5
Hemoglobin subunit epsilon 1 (HBE1)	1	−1.6
	2	−1.5
Hemoglobin subunit gamma 1 (HBG1)	1	−1.6
	2	−1.5
Hemoglobin subunit gamma-2	1	−1.6
	2	−1.6
Heme oxygenase 1 (HMOX1)	1	−1.9
	2	−1.6
Heme oxygenase 2	1	−1.7
	2	−1.6
Solute carrier family 46 member 1 (SLC46A1)	1	−1.8
	2	−1.7
Hydrogen sulfide		
Betaine-homocysteine methyltransferase (BHMT)	1	−0.8
	2	−0.7
Cystathionine beta-synthase (CBS)	1	−0.8
	2	−0.7
Cystathionine γ-lyase	1	−0.7
	2	−0.7
Mercaptopyruvate sulfurtransferase 3 (MPST-3)	1	−0.8
	2	−0.7

## Data Availability

All research data are provided in the current article.

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
