# Peer review of "Modeling the Interactions Between Chemicals and Proteins to Predict the Health Consequences of Air Pollution"

_ijerph, 2025, doi:10.3390/ijerph22030418_

Round 1

Reviewer 1 Report

Comments and Suggestions for Authors

Air pollution is a major threat to human health that impacting globally. Here, the authors investigated the molecular effect of pollutants through the interaction networks, functional and pathways analysis. Therefore, the article is interesting and gives an overview of the interactions between chemicals found in air pollution and proteins within the human body.

Here are some minor comments:

1. Authors are advised to include the research questions and the study's contribution in the introduction.

2.      Given that the authors found that CO and H2S have positive health impacts on people, it is wise to talk about the threshold of exposure that is deemed healthy. Additionally, talk about if the negative impacts outweigh the total positive advantages.

3.      It is best to keep the conclusion brief and free of repetition. 

Author Response

Response to Reviewer 1 Comments

1. Summary

Thank you very much for taking the time to review this manuscript. Please find the detailed responses below and the corresponding corrections highlighted in yellow colour in the re-submitted files.

2. Questions for General Evaluation

Reviewer’s Evaluation

Response and Revisions

Does the introduction provide sufficient background and include all relevant references?

Yes/Can be improved/Must be improved/Not applicable

[Yes]

Are all the cited references relevant to the research?

Yes/Can be improved/Must be improved/Not applicable

[Yes]

Is the research design appropriate?

Yes/Can be improved/Must be improved/Not applicable

[Yes]

Are the methods adequately described?

Yes/Can be improved/Must be improved/Not applicable

[Yes]

Are the results clearly presented?

Yes/Can be improved/Must be improved/Not applicable

[Yes]

Are the conclusions supported by the results?

Yes/Can be improved/Must be improved/Not applicable

[Yes]

3. Point-by-point response to Comments and Suggestions for Authors

Comments 1: [Authors are advised to include the research questions and the study's contribution in the introduction.]

Response 1: [Authors are advised to include the research questions and the study's contribution in the introduction.] Thank you for pointing this out. We agree with this comment. Therefore, we have included research questions and the study’s contribution in the introduction – pages 2-3, lines 72-28, 101-106, and 108-110.

Comments 2: [Given that the authors found that CO and H2S have positive health impacts on people, it is wise to talk about the threshold of exposure that is deemed healthy. Additionally, talk about if the negative impacts outweigh the total positive advantages.]

Response 2: Agree. We have accordingly revised it to emphasize this point. We have found both negative and positive impacts of these airborne chemicals and mentioned the threshold of exposure - pages 4-5 and 10-11, lines 170-171, and 232-233.

Comments 3: [It is best to keep the conclusion brief and free of repetition.]

Response 3: Agree. I have accordingly revised it to emphasize this point. We have elaborated the conclusion and removed repetitive sentences – pages 23-24, lines 468-519.

All the revisions are carefully done and marked by yellow colour.

Reviewer 2 Report

Comments and Suggestions for Authors

Authors utilize the STITCH database to identify interactions between airborne chemical compounds and proteins, and the STRING database to describe the protein-protein interaction (PPI) networks associated with these pollutants. They focused on CO and H2S pollutants and they identified the specific proteins that interact with these two compounds as well as the subsequent signaling network. 

While the study has potential interest, the manuscript is presented in a not well-structured manner and it remains superficial. It does not provide experimental data to substantiate the authors’ speculations. Although the authors pay significant attention to future directions and recommendations, the in-silico approach they used requires experimental validation or support from existing literature. An exhaustive comparison of the findings of this study with available datasets or experimental results, which would significantly make the study more robust, is lacking.

The work would benefit from greater robustness by extending beyond the use of the STITCH and STRING databases. For example, by mapping the networks to identify critical nodes using additional tools of network graph analysis, the protein interactions obtained by STRING can subjected to clustering analysis to identify key pathways. Importantly, the authors should consider using computational tool to predict the hypothetical direct interactions between the airborne chemicals and the proteins identified by STITCH as primary interactors. Algorithms that exploit artificial intelligence, such as  AlphaFold or Rosetta, could be utilized for this purpose. This approach would provide valuable insights into interaction energies between the chemicals and the identified proteins, helping to focus on the most favored interactions for further investigation.

The authors claim to have analyzed various chemicals but they only present results for two of them. It would be helpful to include a table (as supplementary material) summarizing the results for other chemicals to clarify how these two pollutants were selected for detailed study.

The images are low resolution, making it difficult to read the molecule names. Fig 1 is an example. It would be useful to see not only the full protein-protein interaction network but also an analyzed network highlighting the main nodes. 

There are also issues with the references: they are sometimes duplicated or incorrectly cited. For example:

• References [6] and [8] are identical.

• On page 3, lines 110 and 118, the reference is cited as "Osman et al." instead of using numerical notation.

• On page 3, line 125, the reference for the software OmicsBox is missing.

Additionally, a list of abbreviations for the proteins mentioned is absent, which is essential given the large number of molecules discussed in the paper.

Author Response

Response to Reviewer 2 Comments

1. Summary

Thank you very much for taking the time to review this manuscript. Please find the detailed responses below and the corresponding corrections highlighted in green colour in the re-submitted files.

2. Questions for General Evaluation

Reviewer’s Evaluation

Response and Revisions

Does the introduction provide sufficient background and include all relevant references?

Yes/Can be improved/Must be improved/Not applicable

[Can be improved]

Are all the cited references relevant to the research?

Yes/Can be improved/Must be improved/Not applicable

[Must be improved]

Is the research design appropriate?

Yes/Can be improved/Must be improved/Not applicable

[Can be improved]

Are the methods adequately described?

Yes/Can be improved/Must be improved/Not applicable

[Can be improved]

Are the results clearly presented?

Yes/Can be improved/Must be improved/Not applicable

[Must be improved]

Are the conclusions supported by the results?

Yes/Can be improved/Must be improved/Not applicable

[Must be improved]

3. Point-by-point response to Comments and Suggestions for Authors

Comments 1: [An exhaustive comparison of the findings of this study with available datasets or experimental results, which would significantly make the study more robust, is lacking.]

Response 1: [An exhaustive comparison of the findings of this study with available datasets or experimental results, which would significantly make the study more robust, is lacking.] Thank you for pointing this out. We agree with this comment. Therefore, we have compared of our findings with available datasets in the discussion – pages 20-22, lines 352-354, 385-403, 414 -423, and 438-452.

Comments 2: [The work would benefit from greater robustness by extending beyond the use of the STITCH and STRING databases. For example, by mapping the networks to identify critical nodes using additional tools of network graph analysis, the protein interactions obtained by STRING can be subjected to clustering analysis to identify key pathways.]

Response 2: Agree. We have accordingly modified the research design about the identification of critical nodes to emphasize this point. We have identified critical proteins by the Cytohuba plugin of Cytoscape tools. Then, the clustering was constructed by the Enrichr server, which also identified the critical pathways – pages 3, and 12-15, lines 135-142, 249-290. 

Comments 3: [Importantly, the authors should consider using computational tool to predict the hypothetical direct interactions between the airborne chemicals and the proteins identified by STITCH as primary interactors. Algorithms that exploit artificial intelligence, such as AlphaFold or Rosetta, could be utilized for this purpose. This approach would provide valuable insights into interaction energies between the chemicals and the identified proteins, helping to focus on the most favored interactions for further investigation.]

Response 3: Agree. Accordingly, we modified the research design about hypothetical direct interactions by molecular docking to emphasize this point. We have identified CO and H2S-interreacted proteins identified by STITCH from the AlphaFold server. Then, the molecular docking was performed by AutoDock vina and visualized by Discovery Studio software – pages 5, 16-20, lines 145-159, 291-348. 

Comments 4: [The authors claim to have analyzed various chemicals but they only present results for two of them. It would be helpful to include a table (as supplementary material) summarizing the results for other chemicals to clarify how these two pollutants were selected for detailed study.]

Response 4: Agree. Accordingly, we summarized the results of other compounds and submitted them in the supplementary materials.

Comments 5: [The images are low resolution, making it difficult to read the molecule names. Fig 1 is an example. It would be useful to see not only the full protein-protein interaction network but also an analyzed network highlighting the main nodes.]

Response 5: Agree. Accordingly, we have increased the resolutions of all the figures.

Comments 6: [There are also issues with the references: they are sometimes duplicated or incorrectly cited. For example:

• References [6] and [8] are identical.

• On page 3, lines 110 and 118, the reference is cited as "Osman et al." instead of using numerical notation.

• On page 3, line 125, the reference for the software OmicsBox is missing.

Additionally, a list of abbreviations for the proteins mentioned is absent, which is essential given the large number of molecules discussed in the paper.

Response 6: Agree. Accordingly, we have removed the identical references (page 29 and lines 691) and used the numerical notation of “Osman et al.” (page 3 and lines 115), added reference for OmicsBox software (page3 and lines 129), carefully added full forms of identified proteins in the abbreviations paragraph (pages 25-27, lines 579-674).

Additional comments of General Evaluation

Comment 1. Reviewer suggested that the introduction could be improved with valid references.

Response: We have carefully modified the introduction with valid references.

Comment 2. Reviewer suggested that should be added all the relevant references.

Response: We have carefully cited all the valid references relevant to the research topic.

Comment 3. Reviewer suggested that could improve research design.

Response: We have carefully improved the research design.

Comment 4. Reviewer said that the results should be clearly presented.

Response: We have clearly added all identified results to research validation.

Comment 5. Reviewer said that the conclusion should be improved.

Response: We have improved the conclusion and removed repetitive sentences.

Round 2

Reviewer 2 Report

Comments and Suggestions for Authors

The authors answered all the comments almost accurately. Even if some points still need to be refined, especially to make the narrative more fluent, the work has substantially improved.

Comments on the Quality of English Language

The authors should pay attention to sentence construction, sometimes, it is difficult to follow the flow of the text.

Author Response

Comments 1: [The reviewer suggested us to present the results more clearly.]

Response 1: Thank you for pointing this out. We agree with this comment. Therefore, we have tried to clearly present our analyzed data in the results section and have highlighted it in red – pages 4-9, lines 161-236.

Comments 2: [The reviewer suggested us to revise the conclusions to be clearer and meaningful.]

Response 2: Agree. We have revised the conclusions with appropriate information and have highlighted it in red - pages 24-25, lines 490-518.

Response to Comments on the Quality of English Language

  We have tried to improve the quality of English to more clearly express the research.
